# NEPENTHE: Entropy-Based Pruning as a Neural Network Depth's Reducer

## Abstract

While deep neural networks are highly effective at solving complex tasks, their computational demands can hinder their usefulness in real-time applications and with limited-resources systems. Besides, it is a known fact that, for many downstream tasks, off-the-shelf models are over-parametrized. While classical structured pruning can reduce the network's width, the computation's critical path, namely the maximum number of layers encountered at forward propagation, apparently can not be reduced.

In this paper, we aim to reduce the depth of over-parametrized deep neural networks: we propose an e**N**tropy-bas**E**d **P**runing as a n**E**ural **N**etwork dep**TH**'s r**E**ducer (NEPENTHE) to alleviate deep neural networks' computational burden. Based on our theoretical finding, NEPENTHE leverages "unstructured" pruning to bias sparsity enhancement in layers with low entropy to remove them entirely. We validate our approach on popular architectures such as MobileNet, Swin-T and RoBERTa, showing that, when in the overparametrization regime, some layers are linearizable (hence reducing the model's depth) with little to no performance loss. The code will be publicly available upon acceptance of the article.

## 1 Introduction

Artificial Intelligence has undergone a transformative evolution propelled by the advent of Deep Neural Networks (DNNs), which have emerged as instrumental in achieving state-of-the-art outcomes across pivotal computer vision domains, including semantic segmentation Chaudhry et al. (2022), classification Barbano et al. (2022), and object detection Mazzeo et al. (2022). Notably, the pervasive impact of DNNs extends beyond conventional computer vision tasks, showcasing absolute potential in realms such as natural language

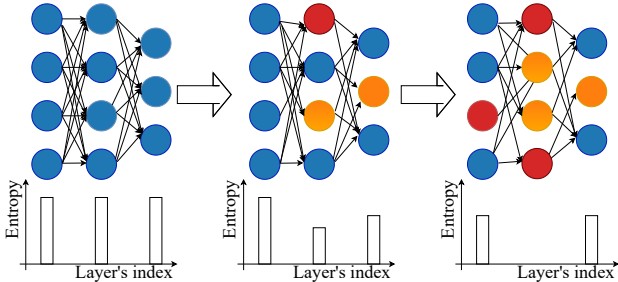

Figure 1: In this work we show that the average neuron's entropy calculated at the layer scale reduces as we induce some sparsity in the model.

processing Touvron et al. (2023), and multi-modal tasks Sun et al. (2019). The employment of DNNs is becoming massive in our lives and looks unstoppable.

While DNNs' performance has exhibited scalability concerning model and dataset size Hestness et al. (2017), the inherent computational burden is one major downside. Notably, contemporary state-of-the-art models are characterized by millions (or even billions) of parameters, demanding billions (or trillions) of floating-point operations (FLOPs) for a single input prediction Guo et al. (2022). Consequently, the substantial resource requirement for training and deploying large neural networks, both in terms of pure hardware capability and energy consumption, poses challenges for real-time applications and edge devices.

Over the past decade, the research landscape has witnessed the emergence of compression techniques as a crucial avenue to address the resource-intensive nature of DNNs. Intrinsically, there ex-

ists a link between the generalization capability of DNNs and the model's complexity: off-the-shelf architectures employed in downstream tasks are, in many cases, over-parametrized, representing a threat for generalization Hestness et al. (2017). One possibility to counter this effect resides in properly removing parameters in excess, providing gains both computation and generalization-wise Han et al. (2015); Tartaglione et al. (2021; 2022). Most of the popular approaches, however, are unable to reduce the number of layers in a DNN.

The impact of removing individual parameters or whole filters on recent computing resources, such as GPUs, is relatively marginal. Due to the parallelization of computations, the size of layers, whether larger or smaller, is primarily constrained by memory caching and core availability. The bottleneck in computation lies in the *critical path* that forward-propagation must traverse Ali Mehmeti-Göpel and Disselhoff (2023), a challenge that can be addressed by strategically removing layers. While some existing works implicitly address this concern Hinton et al. (2015), they fail to guarantee a-priori no performance loss (given that they impose a target shallow model) or avoid substantial perturbations. This motivates the exploration of designing an iterative pruning strategy, aimed at reducing the model's depth while preserving optimal performance.

In this work, we present NEPENTHE, an approach that iteratively attempts to remove layers from a DNN. More specifically, given the large use of *rectifier* activation functions such as ReLU, GELU, and Leaky-ReLU, we can identify the average *state* of a given neuron for the trained task, and from that, we can maximize the utilization of one of the two regions identifiable in these activations by minimizing an entropy. We find that vanilla unstructured pruning is already implicitly minimizing such entropy, but is hardly able to completely make a whole layer utilizing one of these two regions. Through the design of our entropy-weighted pruned parameter budget at the layer's scale, we can favor solutions where the layer's entropy drops to zero, hence becoming linearizable (Fig. 1). We summarize, here below, our key messages and contributions.

- We propose a measure of entropy at the single neuron's scale, which indicates how much such neuron uses its linear part(s): through its minimization, it is in principle possible to linearize it, and by making the average entropy drop to zero, it is possible to linearize the whole layer (Sec. 3.1).
- We theoretically show that "unstructured" pruning, in rectifier-activated layers, naturally reduces the layer's entropy (Sec. 3.1 and Appendix A), validating such result also empirically (Sec. 4.2).
- We propose NEPENTHE, a new method aiming to decrease a neural network's depth (Sec. 3.3) through a proper entropy-guided reweighting of the pruning budget at the layer's scale (Sec. 3.2).
- We test NEPENTHE in a variety of setups and with some popular architectures (Sec. 4.3), showcasing that it can achieve layer removal with little or no performance loss when over-parametrized networks are employed.

## 2 RELATED WORKS

**Neural Network Pruning.** Neural network pruning has gained considerable attention in recent years due to its potential to enhance model performance and reduce over-fitting. Its goal is to reduce a cumbersome network to a smaller one while maintaining accuracy by removing irrelevant weights, filters, or other structures, from neural networks. While *structured* pruning removes entire neurons, filters, or channels Tartaglione et al. (2021); He and Xiao (2023); Lin et al. (2020), *unstructured* pruning algorithms remove weights without explicitly considering the neural network's structure Han et al. (2015). Magnitude-based pruning, where the importance score to prune parameters is based on their magnitude Han et al. (2015); Louizos et al. (2018); Zhu and Gupta (2017), and gradient-based pruning, where the ranking or the penalty term is a function of the gradient magnitude (or to higher order derivatives) Lee et al. (2019); Tartaglione et al. (2022), are the main types of unstructured pruning approaches. Blalock et al. (2020) compared the effectiveness of these approaches and concluded that, in general, gradient-based methods are less accurate than magnitude-based methods. Moreover, Gale et al. (2019) showed that simple magnitude pruning approaches achieve comparable or better results than complex methods, making them a good trade-off between complexity and competitiveness. Computationally-wise, it is broadly known that, in general-purpose hardware setup, structured pruning can provide larger benefits, in terms of both

memory and computation, than unstructured approaches, despite the achieved sparsity rate can be substantially lower Bragagnolo et al. (2021).

**Entropy-Guided Pruning.** Some works have already tried to propose entropy-based approaches to guide pruning. For convolutional neural networks, Luo and Wu (2017) put forward an iterative filter pruning strategy in which the importance of each filter is calculated by their entropy-based channel selection metric. To recover performance, the pruned model is then fine-tuned. Also for CNNs, Hur and Kang (2019) suggested an entropy-based method that determines dynamically during training the threshold by considering the average amount of information from the weights to output. Moreover, Min et al. (2018) proposed a two-stage filter pruning framework, first intra-layer and then extra-layer. Given that the entropy is a measure of disorder, evidently, it identifies filters that mutually have low entropy: these can be considered *redundant* and for instance, can be removed from the model. These approaches, despite reducing the layer's width, are not designed to tackle our aim: removing entire layers to reduce DNNs' depth.

**Neural Network Depth Reduction.** Towards neural network depth reduction, Chen and Zhao (2019) inspect the possibility of having a layer-wise pruning method based on feature representation, a-posteriori employing a retraining strategy that utilizes knowledge distillation. This work reinforces the possibility of designing a layer-pruning algorithm. Endorsing this, Dror et al. (2022) proposed a method that learns whether non-linear activations can be removed, allowing the folding of consecutive linear layers into one. More specifically, ReLU-activated layers are replaced with PReLU activations, showcasing a regularized slope. Post-training, the PReLUs almost linear are removed, and the layer can be folded with its subsequent one. Ali Mehmeti-Göpel and Disselhoff (2023) proposes a similar channel-wise approach that enables reducing more non-linear units in the network while maintaining similar performance. While these works sought to shrink the neural network's depth by working at the activation level and forcing it to stay either linear or non-linear, our approach does not directly enforce any of that. In rectifier-activated networks, we perform a targeted unstructured pruning that off-line favors either the neuron's shutdown or the use of its linear part.

By prioritizing pruning connections in low-entropy layers, Liao et al. (2023) also develop an unstructured entropy-guided pruning method which reduces DNNs' depth. Nevertheless, EGP only enables the removal of a limited number of layers, notwithstanding its effectiveness. Indeed, the accuracy drastically decreases when several layers are removed. This will be confirmed in Sec. 4 by contrasting this approach with NEPENTHE.

## 3 NEPENTHE

In this section, we present our method NEPENTHE, which focuses on pruning connections in layers with low entropy to remove them entirely. First, we show that unstructured pruning naturally minimizes the neuron's entropy (in rectifier-activated layers). This will motivate our entropy-guided pruning approach, which allows a gradual layer removal.

### 3.1 ENTROPY FOR RECTIFIER ACTIVATIONS

Let us assume $\psi$ is the rectifier of the $l$-th layer, populated by $N_l$ neurons. We can monitor the output $y_{l,i}^{\boldsymbol{x}}$ of the $i$-th neuron from a given input $\boldsymbol{x}$ of the dataset $\mathcal{D}$ and write it as:

$$y_{l,i}^{\boldsymbol{x}} = \psi(z_{l,i}^{\boldsymbol{x}}), \tag{1}$$

where $z_{l,i}^{\boldsymbol{x}}$ is the output of the $i$-th neuron inside the $l$-th layer. From equation 1, we can define three possible "states" for the neuron:

$$s_{l,i}^{\boldsymbol{x}} = \begin{cases} +1 & \text{if } y_{l,i}^{\boldsymbol{x}} > 0 \\ -1 & \text{if } y_{l,i}^{\boldsymbol{x}} < 0 \\ 0 & \text{if } y_{l,i}^{\boldsymbol{x}} = 0 \end{cases} \tag{2}$$

More synthetically, for the output of the $i$-th neuron, we can easily identify in which of these states we are by simply applying the sign function to $z_{l,i}^{\boldsymbol{x}}$, obtaining $s_{l,i}^{\boldsymbol{x}} = \text{sign}(z_{l,i}^{\boldsymbol{x}})$. Informally, we can say that the neuron is in the *ON State* when $s_{l,i}^{\boldsymbol{x}} = +1$ (as it is typically the linear region) while it

is in the *OFF State* when $s_{l,i}^{\boldsymbol{x}} = -1$ (given that $\lim_{x \to -\infty} \psi(x) = 0$).[1] The third State $s_{l,i}^{\boldsymbol{x}} = 0$ is a special case, as it can be either mapped as an ON or OFF State. From the average over a batch of outputs for the neuron, we can obtain the probability (in the frequentist sense) of the i-th neuron of being in either the ON or the OFF States. For instance, we can obtain the probability of the ON State as:

$$p(s_{l,i}\!=\!+1) = \begin{cases} \frac{1}{S_{l,i}} \sum_{j=1}^{\|\mathcal{D}\|_0} s_{l,i}^{\boldsymbol{x}_j} \Theta(s_{l,i}^{\boldsymbol{x}_j}) & \text{if } S_{l,i} \neq 0 \\ 0 & \text{otherwise,} \end{cases} \tag{3}$$

where

$$S_{l,i} = \sum_{j=1}^{\|\mathcal{D}\|_0} \left| s_{l,i}^{\boldsymbol{x}_j} \right| \tag{4}$$

counts how many times the ON and the OFF states are encountered, $\|\mathcal{D}\|_0$ is the number of the input samples, and $\Theta$ is the Heaviside function.[2] Evidently, we exclude the third state from this count as it can be associated with being either within ON or OFF. Given that we are either interested in the ON or the OFF States, we can then deduce that, when $S_{l,i} \neq 0$, $p(s_{l,i}\!=\!-1) = 1 - p(s_{l,i}\!=\!+1)$. Given this, we can calculate the entropy of the i-th neuron in the l-th layer as follows:

$$\mathcal{H}_{l,i} = - \sum_{s_{l,i}=\pm 1} p(s_{l,i}) \log_2 [p(s_{l,i})] \tag{5}$$

With the definition in equation 5, $\mathcal{H}_{l,i}$ can be zero in two possible cases:

- $s_{l,i} = -1 \ \forall j$. In this case, $z_{l,i} \leq 0 \ \forall j$. When employing a ReLU, the output of the i-th neuron is always 0, and in this specific case, the neuron can be simply pruned.
- $s_{l,i} = +1 \ \forall j$. In this case, $z_{l,i} \geq 0 \ \forall j$. The output of the i-th neuron is always the same as its input,[3] this neuron can in principle be absorbed by the following layer as there is no non-linearity between them anymore.

By averaging the entropy values for the total number of neurons $N_l$ inside the l-th layer, we can define the average entropy of the l-th layer as:

$$\widehat{\mathcal{H}}_l = \frac{1}{N_l} \sum_i \mathcal{H}_{l,i}. \tag{6}$$

Since we aim to minimize the depth of deep neural networks by eliminating zero-entropy layers, we would like to have $\widehat{\mathcal{H}}_l = 0$. Unfortunately, directly minimizing equation 6 in the optimization function is hard as it relies on non-differentiable measures like equation 3. However, unstructured pruning can surprisingly be a promising choice for such a goal.

Under the assumptions of having weights distributed according to a Gaussian $f_W$, after applying a threshold $t$ on their magnitude, their distribution will become $f_{\widehat{W}}$ (Fig. 2a). Assuming as well that the input is Gaussian, we will have the post-synaptic potential distribution $f_Z$ as pictured in Fig. 2b. Its dependence on the threshold parameter allows us to also derive the entropy as a function of $t$ (Fig. 2c): as we observe, the entropy decreases given that the threshold increases: through unstructured pruning, the neuron's output entropy is naturally minimized when employing rectified activations, even in the oversimplified case here treated. The derivation leading to the results in Fig. 2 is provided in Appendix A , and the Gaussian assumption is validated empirically in Appendix B.

In the following, we will present how we are exploiting such a property of unstructured pruning towards layer entropy minimization.

## 3.2 A Layer Entropy-Aware Pruning Score

Driven by the promising theoretical results presented in Sec. 3.1 and Appendix A, we will design here a relevant metric that will guide the unstructured pruning to lower the whole layer's entropy

---

[1]There are few exceptions, such as LeakyReLU. In these cases, although the activation doesn't converge to zero, we still call it the OFF state since the output's magnitude is lower for the same input magnitude.

[2]For convolutional layers, it is necessary to sum and average over the entire feature map generated per input.

[3]or very close as in GeLU.

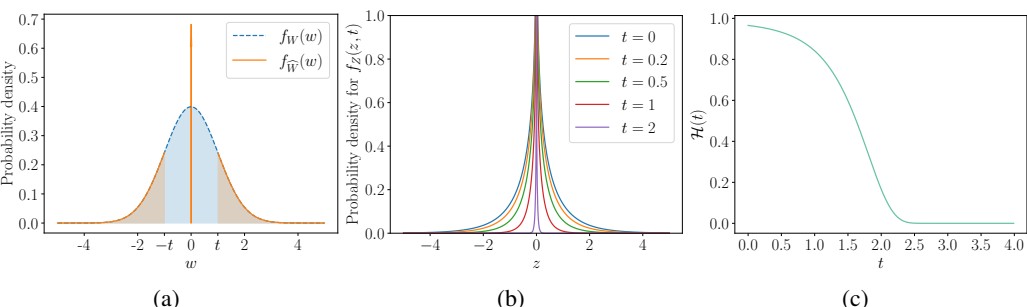

(a)                                    (b)                                    (c)

Figure 2: Distribution of a layer's parameters with magnitude pruning at threshold $t$ (a); pre-activation distribution at varying $t$ under the assumption of independence and centering of the Gaussian distributed input and layer's parameters (b); entropy of the rectifier-activated neuron's output as a function of $t$ (c), all in the large $N$ limit.

$\widehat{\mathcal{H}}_l$. As we aim to increase the number of zero-entropy layers, intuitively more pruning should be applied to layers with lower entropy, as they are the best candidates to be removed. Concurrently, to minimize the impact on performance, only low-magnitude weights should be removed, as they are typically those providing the lowest contribution to the neural network's output Han et al. (2015); Tartaglione et al. (2021). To reach these two objectives, we first define an intra-layer's pruning irrelevance score

$$\mathcal{I}_l = \frac{1}{N_l} \sum_{i=1}^{N_l} \widehat{\mathcal{H}}_{l,i} \cdot \frac{1}{\|\boldsymbol{w}_l\|_0} |w_{l,i}|, \tag{7}$$

where $\|\boldsymbol{w}_l\|_0$ is the current layer's parameters cardinality (hence, not accounting for the already pruned weights, if any). This metric accounts for the average parameter's magnitude and the layer's entropy at the same time: layers with few parameters but high entropy are less prone to be removed than layers with more parameters but lower entropy (under the same parameter's norm constraint). Besides, the parameter's magnitude of neurons with zero entropy is not accounted for in the importance score calculation. Symmetrically, to remove parameters from layers having lower pruning irrelevance, we define the inter-layer's pruning relevance score $\mathcal{R}_l$ as:

$$\mathcal{R}_l = \begin{cases} \frac{1}{\mathcal{I}_l} \sum_{j \in L} \mathcal{I}_j & \text{if } \mathcal{I}_l \neq 0 \\ 0 & \text{otherwise.} \end{cases} \tag{8}$$

This measure is as large as the $l$-th layer's pruning irrelevance score is smaller compared to the other layer's. Noticeably, $\mathcal{R}_l \in [1; +\infty)$: to exactly establish how many parameters $\|\boldsymbol{w}_l\|_0^{\text{pruned}}$ should be removed inside each layer $l$ at a given pruning iteration, we have the *entropy-weighted pruned parameter budget*

$$\|\boldsymbol{w}_l\|_0^{\text{pruned}} = \|\boldsymbol{w}\|_0^{\text{pruned}} \cdot \frac{\exp[\mathcal{R}_l]}{\sum_j \exp[\mathcal{R}(j)]}. \tag{9}$$

Here follows an overview of NEPENTHE.

### 3.3 Entropy-Based Iterative Pruning

Depicted in Alg. 1,[4] we guide our entropy-based iterative pruning algorithm to remove layers with zero entropy. Indeed, if a layer has an entropy equal to zero, then all of its neurons have an entropy equal to zero: $\widehat{\mathcal{H}}_l = 0 \Leftrightarrow \mathcal{H}_{l,i} = 0 \,, \forall i$. Hence, this layer doesn't necessarily need to have a rectifier: this layer can be removed entirely without the need for future pruning. Towards this end, we first train the neural network, represented by its weights at initialization $\boldsymbol{w}^{\text{init}}$, on the training set $\mathcal{D}_{\text{train}}$ (line 2) and evaluate it on the validation set $\mathcal{D}_{\text{val}}$ (line 3). As defined in equation 6, we then calculate the entropy $\widehat{\mathcal{H}}$ on the training set $\mathcal{D}_{\text{train}}$ for each layer $l$ of the considered list of layers $L$ (line 6). This list is initialized to all the layers of the neural network having a rectifier activation (hence, the output layer is excluded).
Considering that $\zeta$ represents the percentage of parameters to remove at each pruning iteration and

---

[4]the function `Weights_to_prune` is presented in the Appendix C.

---

**Algorithm 1** Our proposed method NEPENTHE.

---

1: **function** NEPENTHE($\boldsymbol{w}^{\text{INIT}}$, $L$, $\mathcal{D}$, $\zeta$, $\theta$)
2: $\quad \boldsymbol{w} \leftarrow \text{Train}(\boldsymbol{w}^{\text{init}}, \mathcal{D}_{\text{train}})$
3: $\quad \text{dense\_acc} \leftarrow \text{Evaluate}(\boldsymbol{w}, \mathcal{D}_{\text{val}})$
4: $\quad \text{current\_acc} \leftarrow \text{dense\_acc}$
5: $\quad$ **while** $\text{current\_acc} > \theta \cdot \text{dense\_acc}$ **do**
6: $\qquad \widehat{\mathcal{H}} \leftarrow \text{Entropy}(\boldsymbol{w}, L, \mathcal{D}_{\text{train}})$
7: $\qquad \|\boldsymbol{w}\|_0^{\text{pruned}} \leftarrow \zeta \cdot \|\boldsymbol{w}\|_0$
8: $\qquad \|\boldsymbol{w}_L\|_0^{\text{pruned}} \leftarrow \text{Weights\_to\_prune}(L, \widehat{\mathcal{H}}, \|\boldsymbol{w}\|_0^{\text{pruned}}, \mathcal{D}_{\text{train}})$
9: $\qquad \boldsymbol{w} \leftarrow \text{Prune}(\|\boldsymbol{w}_L\|_0^{\text{pruned}})$
10: $\qquad \boldsymbol{w} \leftarrow \text{Train}(\boldsymbol{w}, \mathcal{D}_{\text{train}})$
11: $\qquad \text{current\_acc} \leftarrow \text{Evaluate}(\boldsymbol{w}, \mathcal{D}_{\text{val}})$
12: $\quad$ **end while**
13: $\quad$ **return** $\boldsymbol{w}$
14: **end function**

---

$\|\boldsymbol{w}\|_0$ the total weight parameters of the considered $L$ layers in the model, we can define the number of weight parameters to be pruned at each iteration $\|\boldsymbol{w}\|_0^{\text{pruned}}$ (line 7) as:

$$\|\boldsymbol{w}\|_0^{\text{pruned}} = \zeta \cdot \|\boldsymbol{w}\|_0. \tag{10}$$

To determine the parameters to prune in each layer, we define a function Weights\_to\_prune. This function calculates the weights to remove for each layer and returns a list indicating the number of neurons that need to be removed from each layer, as discussed in Sec. 3.2. At this point, for each layer $l$, the neurons having non-zero entropy are first selected and then $\|\boldsymbol{w}_l\|_0^{\text{pruned}}$ non-zero weights having the lowest absolute magnitude are removed (line 9). The model is then retrained (line 10) and re-evaluated on the validation set $\mathcal{D}_{\text{val}}$ (line 11). The final model is obtained once the performance on the validation set drops below some relative threshold $\theta$.

## 4 EXPERIMENTS

In this section, we empirically evaluate the effectiveness of our proposed approach, NEPENTHE, across multiple architectures and datasets for traditional image classification and natural language processing setups. We compare our results with the iterative magnitude pruning (IMP) method from Han et al. (2015). Additionally, in image classification tasks, we compare our results with two other approaches: removing the layers having the lowest sum weights/gradients. Then, we also induce sparsity inside layers with Hrank Lin et al. (2020), a filter pruning method which removes filters with low-rank feature maps. Then, we also minimize the group lasso penalty for each layer using the method outlined in Ochiai et al. (2017). We also compare our results with the existing approaches: EGP Liao et al. (2023) and Layer Folding Dror et al. (2022), both effectively removing layers for image classification models.

### 4.1 EXPERIMENTAL SETUP

A variety of setups is covered by evaluating our method on three popular models: ResNet-18 He et al. (2016), MobileNet-V2 Howard et al. (2017), and Swin-T Liu et al. (2021), trained on five datasets: CIFAR-10 Krizhevsky et al. (2009), Tiny-ImageNet Le and Yang (2015), and PACS, VLCS, and SVIRO from DomainBed Gulrajani and Lopez-Paz (2020), following the same training policies as Quétu and Tartaglione (2024) and Xu et al. (2021). Moreover, two natural language processing models: BERT Kenton and Toutanova (2019) and RoBERTa Liu et al. (2019) are trained on three datasets: SST-2 Socher et al. (2013), QNLI Williams et al. (2018), and RTE Bentivogli et al. (2009), with the training strategies of Peer et al. (2022). In all the setups, we set $\zeta = 0.5$ for ResNet-18, $\zeta = 0.25$ for Swin-T, and $\zeta = 0.1$ for MobileNet-V2. Moreover, we set $\zeta = 0.25$ (respectively $\zeta = 0.15$) for the models trained on QNLI and RTE (respectively SST-2). The results of Layer Folding (respectively EGP) are obtained using the same aforementioned training policy, with the hyper-parameters declared in Dror et al. (2022) (respectively in Liao et al. (2023)). All the

Table 1: Trend in the bottom six layer's entropies for ResNet-18 trained on CIFAR-10.

| Approach | $\widehat{\mathcal{H}}_1$ | $\widehat{\mathcal{H}}_2$ | $\widehat{\mathcal{H}}_3$ | $\widehat{\mathcal{H}}_4$ | $\widehat{\mathcal{H}}_5$ | $\widehat{\mathcal{H}}_6$ | top-1 |
|---|---|---|---|---|---|---|---|
| Dense | 0.647 | 0.680 | 0.728 | 0.785 | 0.791 | 0.797 | 91.66 |
| IMP (iter #1) | 0.585 | 0.650 | 0.699 | 0.725 | 0.767 | 0.778 | 92.29 |
| IMP (iter #2) | 0.506 | 0.580 | 0.647 | 0.654 | 0.700 | 0.722 | 92.25 |
| IMP (iter #3) | 0.256 | 0.623 | 0.658 | 0.672 | 0.682 | 0.737 | 92.46 |
| IMP (iter #4) | 0.192 | 0.660 | 0.667 | 0.676 | 0.698 | 0.763 | 92.27 |
| IMP (iter #5) | 0.136 | 0.589 | 0.648 | 0.727 | 0.728 | 0.791 | 92.44 |
| IMP (iter #6) | 0.093 | 0.447 | 0.640 | 0.650 | 0.764 | 0.765 | 91.89 |
| IMP (iter #7) | 0.055 | 0.335 | 0.487 | 0.592 | 0.640 | 0.775 | 91.66 |
| NEPENTHE | 0 | 0 | 0 | 0.014 | 0.121 | 0.942 | **92.55** |

hyperparameters, augmentation strategies, learning policies, and how we choose $\zeta$ are provided in the Appendix C. We also implement our method for ResNet-50, ResNet-152, MobileNetV2-0.75 models trained on CIFAR-10 and ResNet-18 trained on Imagenet Deng et al. (2009). The results of these setups are shown in Appendix D.2.

## 4.2 TREND OF LAYER'S ENTROPY

As a preliminary experiment, we will study here the effect of pruning on the layer's entropy. Table 1 reports the entropy trend of the six layers showing the lowest entropy.

The iterative magnitude approach removes progressively, in this setup, the 50% of the parameters from the model, following a vanilla global unstructured magnitude pruning approach. As expected from the derivation as in Sec. 3.1, as the pruning progresses (and implicitly $t$ grows), the entropy is naturally decreased, showcasing very small values after some pruning iterations. However, we also observe that as the entropy $\widehat{H}_1$ decreases, the top-1 accuracy begins to deteriorate. This happens as there is no proper pruning re-allocation, that instead happens with NEPENTHE according to equation 8: indeed, in such case not only does the performance remain high, but we can successfully remove three layers from the model.

Noticeably, $\widehat{H}_4$ and $\widehat{H}_5$ are also very low, while already starting from $\widehat{H}_6$ the entropy is very high. Contrarily to magnitude pruning where the entropy is in general in intermediate-range values, NEPENTHE tries to push all the encoded information toward layers having already high entropy, enabling effective layer removal with little (or in this case no) performance loss. This is also illustrated in Fig. 3, showing the distribution of the neuron states per layer for ResNet-18 on CIFAR-10 trained with NEPENTHE. Our unstructured pruning approach effectively removes three layers by pushing all the neurons inside low-

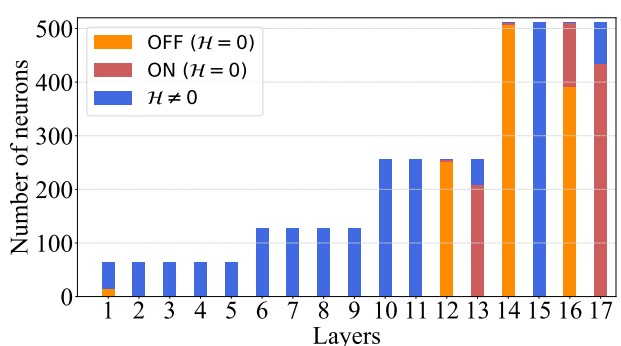

Figure 3: Neuron states per layer for ResNet-18 trained on CIFAR-10 pruned by NEPENTHE.

entropy layers to be either in the ON or in the OFF state. Besides, we also notice that in some layers (like 13 and 17) there are entire units at zero entropy- we also achieve some structured sparsity by an unstructured approach, as already reported in some works Han et al. (2015); Tartaglione et al. (2021).

We also analyzed the layers pruned from a ResNet-18 trained on CIFAR-10. We observed differences in the layers pruned by our entropy-based method compared to other pruning methods. Indeed, with our method, the layers with the lowest entropy are typically found near the deepest layers of the network. Similarly, in layer folding and EGP methods, the layers near the output are

Table 2: Test performance (top-1), lowest non-zero layers' entropy ($\widehat{\mathcal{H}}_{min}$) and the number of removed layers (Rem.) for all the considered image classification setups. The results achieved by our method are in italic.

| Dataset | Approach | ResNet-18 | | | MobileNet-V2 | | | Swin-T | | |
|---|---|---|---|---|---|---|---|---|---|---|
| | | $\widehat{\mathcal{H}}_{min}$ | top-1 | Rem. | $\widehat{\mathcal{H}}_{min}$ | top-1 | Rem. | $\widehat{\mathcal{H}}_{min}$ | top-1 | Rem. |
| CIFAR-10 | Dense model | 0.647 | 91.66 | 0/17 | 0.386 | 93.68 | 0/35 | 0.028 | 91.54 | 0/12 |
| | Smallest weights | 0.582 | 10.00 | 1/17 | 0 | 10.00 | 1/35 | 0.031 | 89.22 | 2/12 |
| | Smallest gradients | 0.314 | 9.29 | 1/17 | 0 | 10.00 | 1/35 | 0.03 | 89.21 | 2/12 |
| | Hrank | 0.001 | 91.70 | 0/17 | 0.055 | 91.73 | 0/35 | 0.048 | 91.87 | 0/12 |
| | Group lasso | 0.130 | 92.11 | 1/17 | 0.001 | 83.00 | 4/35 | 0.028 | 91.68 | 0/12 |
| | IMP | 0.055 | 91.66 | 0/17 | 0.046 | **93.50** | 0/35 | 0.286 | 90.53 | 0/12 |
| | EGP | - | 92.18 | 3/17 | - | 92.22 | 6/35 | - | 92.01 | 1/12 |
| | Layer folding | - | 90.65 | 1/17 | - | 87.84 | 6/35 | - | 85.73 | 2/12 |
| | *NEPENTHE* | *0.121* | ***92.55*** | ***3/17*** | *0.001* | *93.26* | *7/35* | *0.362* | ***92.29*** | *2/12* |
| Tiny-ImageNet | Dense model | 0.471 | 41.44 | 0/17 | 0.076 | 45.86 | 0/35 | 0.067 | 75.60 | 0/12 |
| | Smallest weights | 0 | 0.5 | 1/17 | 0 | 0.5 | 1/35 | 0.07 | **75.12** | **1/12** |
| | Smallest gradients | 0 | 0.5 | 1/17 | 0.099 | 46.62 | 1/35 | 0.07 | 74.54 | 1/12 |
| | Group lasso | 0.418 | **41.92** | 0/17 | 0.001 | 47.1 | 0/35 | 0.124 | 71.3 | 0/12 |
| | IMP | 0.464 | 39.14 | 0/17 | 0.013 | 45.24 | 0/35 | 0.104 | 67.56 | 0/12 |
| | EGP | - | 39.50 | 4/17 | - | 47.52 | 6/35 | - | 71.48 | 1/12 |
| | Layer folding | - | 37.86 | 4/17 | - | 25.88 | 12/35 | - | 50.54 | 1/12 |
| | *NEPENTHE* | *0.129* | *39.56* | ***5/17*** | *0.002* | ***47.92*** | ***12/35*** | *0.126* | *72.58* | *1/12* |
| PACS | Dense model | 0.332 | 94.70 | 0/17 | 0.207 | 93.20 | 0/35 | 0.057 | 97.10 | 0/12 |
| | Smallest weights | 0.122 | 16.20 | 1/17 | 0 | 18.5 | 1/35 | 0.078 | 96.1 | **2/12** |
| | Smallest gradients | 0.115 | 16.20 | 1/17 | 0.063 | 16.2 | 1/35 | 0.069 | 95.7 | 2/12 |
| | Group lasso | 0.831 | 81.2 | 0/17 | 0.176 | 95.10 | 0/35 | 0.063 | **96.3** | 0/12 |
| | IMP | 0.280 | **90.80** | 0/17 | 0.170 | **95.40** | 0/35 | 0.101 | 93.90 | 0/12 |
| | EGP | - | 84.30 | 2/17 | - | 17.7 | 3/35 | - | 93.5 | 1/12 |
| | Layer folding | - | 82.90 | 3/17 | - | 79.70 | 1/35 | - | 87.70 | 2/12 |
| | *NEPENTHE* | *0.030* | *90.10* | ***3/17*** | *0.080* | *92.20* | ***1/35*** | *0.335* | *95.10* | *2/12* |
| VLCS | Dense model | 0.382 | 80.89 | 0/17 | 0.258 | 81.83 | 0/35 | 0.070 | 86.58 | 0/12 |
| | Smallest weights | 0.122 | 46.13 | 1/17 | 0 | 6.43 | 1/35 | 0.063 | 84.62 | 1/12 |
| | Smallest gradients | 0.122 | 46.13 | 1/17 | 0.176 | 46.13 | 1/35 | 0.064 | 84.15 | 1/12 |
| | Group lasso | 0.831 | 67.85 | 0/17 | 0.176 | 78.84 | 0/35 | 0.063 | 84.81 | 0/12 |
| | IMP | 0.357 | 74.09 | 0/17 | 0.273 | **80.43** | 0/35 | 0.139 | 80.06 | 0/12 |
| | EGP | - | 74.28 | 2/12 | - | 45.85 | 2/35 | - | 82.95 | 1/12 |
| | Layer folding | - | 64.87 | 1/17 | - | 68.87 | 2/35 | - | 70.92 | 1/12 |
| | *NEPENTHE* | *0.224* | ***78.38*** | ***2/17*** | *0.001* | *80.06* | ***2/35*** | *0.411* | ***85.27*** | ***1/12*** |
| SVIRO | Dense model | 0.336 | 99.93 | 0/17 | 0.187 | 99.95 | 0/35 | 0.060 | 99.95 | 0/12 |
| | Smallest weights | 0.122 | 35.55 | 1/12 | 0.014 | 35.55 | 1/35 | 0.039 | 99.70 | 4/12 |
| | Smallest gradients | 0.122 | 35.55 | 1/12 | 0.014 | 35.55 | 1/35 | 0.0154 | 99.55 | 4/12 |
| | Group lasso | 0.803 | 99.77 | 0/12 | 0.795 | 99.93 | 0/35 | 0.014 | **99.79** | 0/35 |
| | IMP | 0.308 | **99.95** | 0/17 | 0.146 | 99.95 | 0/35 | 0.260 | 99.75 | 0/12 |
| | EGP | - | 99.88 | 5/17 | - | 35.05 | 2/35 | - | 99.64 | 5/12 |
| | Layer folding | - | 99.46 | 8/17 | - | 99.83 | 2/35 | - | 99.66 | 5/12 |
| | *NEPENTHE* | *0.001* | *99.61* | ***8/17*** | *0.020* | ***99.98*** | ***2/35*** | *0.162* | *99.75* | ***5/12*** |

often pruned first. This is because their nonlinear activations have minimal robustness, making them suitable candidates for pruning. In contrast, methods that prune layers based on the lowest sum of weights/gradients tend to remove layers near the input of the model first. These layers usually have fewer parameters and, thus, a lower cumulative weight or gradient sum. As a result, they are identified as less important by these pruning criteria. The visualization of layer removal by different methods is presented in Fig. 14, Appendix D.1.

Here follows an extensive analysis of more datasets and architectures.

## 4.3 RESULTS

**Image classification tasks.** Table 2 shows the test performance (top-1), the lowest non-zero layer's entropy ($\widehat{\mathcal{H}}_{min}$) as well as the number of removed layers (Rem.) for all the considered image classification setups. Since Layer Folding is changing the architecture by hand, it is inconvenient to calculate $\widehat{\mathcal{H}}_{min}$. Moreover, the entropy in EGP Liao et al. (2023) is calculated differently: the $\widehat{\mathcal{H}}_{min}$ for models obtained with EGP are hence omitted in the table to avoid confusion. It appears that removing layers with the lowest sum weights/gradients is very effective with Swin-T. However, after removing one layer by applying these methods on ResNet-18 and MobileNet-V2, the models'

Table 3: Test performance (top-1), lowest non-zero layers' entropy ($\widehat{\mathcal{H}}_{min}$) and the number of removed layers (Rem.) for all the considered NLP setups.

| Dataset | Approach | BERT | | | RoBERTa | | |
|---|---|---|---|---|---|---|---|
| | | $\widehat{\mathcal{H}}_{min}$ | top-1 | Rem. | $\widehat{\mathcal{H}}_{min}$ | top-1 | Rem. |
| QNLI | Dense model | 0.173 | 90.48 | 0/12 | 0.190 | 92.18 | 0/12 |
| | IMP | 0.307 | 85.87 | 0/12 | 0.263 | **89.04** | 0/12 |
| | *NEPENTHE* | *0.251* | **88.69** | **4/12** | *0.001* | 87.41 | **2/12** |
| RTE | Dense model | 0.211 | 61.01 | 0/12 | 0.236 | 66.79 | 0/12 |
| | IMP | 0.335 | 57.76 | 0/12 | 0.314 | 62.82 | 0/12 |
| | *NEPENTHE* | *0.001* | **58.12** | **4/12** | *0.001* | **66.06** | **1/12** |
| SST-2 | Dense model | 0.114 | 92.20 | 0/12 | 0.131 | 92.66 | 0/12 |
| | IMP | 0.301 | 88.65 | 0/12 | 0.125 | **91.51** | 0/12 |
| | *NEPENTHE* | *0.001* | **88.99** | **3/12** | *0.001* | 89.79 | **4/12** |

Table 4: Ablation study on ResNet-18 trained on CIFAR-10. Each component contributes to the effectiveness of NEPENTHE.

Table 5: MFLOPs, Inference time [ms], Memory usage [MBs] and Energy consumption [mJ] of ResNet-18 on CIFAR-10 on NVIDIA A4500.

| Entropy | Don't care state | Neurons Selection | top-1 | Rem. |
|---|---|---|---|---|
| | | | 91.66 | 0/17 |
| ✓ | | | 92.18 | 3/17 |
| ✓ | ✓ | | 92.33 | 3/17 |
| ✓ | ✓ | ✓ | 92.55 | 3/17 |

| Rem. | MFLOPs | Inference time [ms] | Mem.usage [MBs] | Energy [mJ] | top-1 |
|---|---|---|---|---|---|
| 0/17 | 725.47 | 3.32 | 230 | 498.7 | 91.66 |
| 1/17 | 258.24 | 3.27 | 202 | 490.2 | 92.25 |
| 3/17 | 231.79 | 2.96 | 170 | 444.0 | 92.55 |
| 5/17 | 159.05 | 2.60 | 60 | 389.7 | 89.30 |

performances degrade to the level of random guesses. Since Hrank operates at the level of the neuron, even though it can help models maintain a good (or even better) performance after pruning, no layer can be removed with this method. Therefore, in order to save computational resources, we only perform this method on CIFAR-10. Also, although minimizing the group lasso penalty has little impact on the performance, its effectiveness in layer removal is not significant. The IMP approach, although not leading to significant performance degradation, does not support the removal of any layers. Conversely, Layer Folding and EGP enable the removal of some layers but at the expense of compromising generalizability. In contrast, NEPENTHE produces models with a substantial number of removable layers with little (or no) performance loss with respect to the dense model's performance. It is also noticeable that in most cases, compared to Layer Folding and EGP, NEPENTHE yields better results, either better top-1 accuracy, more removable layers, or both.

**NLP tasks.** The results for all NLP setups are presented in Table 3. Similarly to what was observed for image classification setups, we observe that while the IMP method does not significantly harm performance, it does not support whole-layer removal, despite minimizing the layer's entropy. In contrast, NEPENTHE produces models with a significant number of removable layers while maintaining a performance comparable to the dense models.

## 4.4 ABLATION STUDY

We will perform, in this section, several different studies: the first is a classical ablation, where we analyze the contribution of each term employed within NEPENTHE, and the second where we will test NEPENTHE with some of the most popular rectifiers. Finally, we showcase the energy-saving and efficiency improvement imposed by NEPENTHE.

Table 4 provides an ablation study on the three key components identifiable within NEPENTHE: the entropy-based weighted pruned parameter budget equation 8, the presence of the don't care state in the entropy formulation equation 2 and the filtering mechanism of non-zero entropy neurons equation 7. Every component contributes towards the effectiveness of NEPENTHE.

Table 6 shows the test performance of ResNet-18 on CIFAR-10, for different rectifiers. NEPENTHE is not dependent on any particular rectifier and can be effective with any since our method removes three layers without performance loss for all the tested activations.

Table 6: Different activation functions on ResNet-18 trained on CIFAR-10.

| Activation | Method | top-1 | Rem. |
|---|---|---|---|
| ReLU | Dense | 91.66 | 0/17 |
| | NEPENTHE | 92.55 | 3/17 |
| SiLU | Dense | 91.66 | 0/17 |
| | NEPENTHE | 92.77 | 3/17 |
| PReLU | Dense | 91.25 | 0/17 |
| | NEPENTHE | 92.27 | 3/17 |
| LeakyReLU | Dense | 91.66 | 0/17 |
| | NEPENTHE | 92.49 | 3/17 |
| GELU | Dense | 91.89 | 0/17 |
| | NEPENTHE | 92.57 | 3/17 |

Table 7: Test performance (top-1) and the number of removed layers (Rem.) for models trained on CIFAR-10 dataset and pruned by the NEPENTHE-finetuning method.

| Model | Approach | top-1 | Rem. |
|---|---|---|---|
| ResNet-18 | Dense model | 91.66 | 0/17 |
| | NEPENTHE-finetuning | 91.63 | 1/17 |
| Swin-T | Dense model | 91.54 | 0/12 |
| | NEPENTHE-finetuning | 86.93 | 1/12 |
| MobileNet-V2 | Dense model | 93.68 | 0/35 |
| | NEPENTHE-finetuning | 93.08 | 6/35 |

Finally, Table 5 showcases the potential savings in terms of FLOPS and inference time on an NVIDIA A4500 GPU for a ResNet-18 trained on CIFAR-10 with NEPENTHE: the fewer layers the network has, the shorter the inference time and the smaller the number of FLOPs.

## 4.5 LIMITATIONS AND FUTURE WORK

NEPENTHE is a successful approach to alleviate deep neural networks' computational burden by decreasing their depth. Nevertheless, this method also presents some limits.

Due to its iterative nature, NEPENTHE leads to longer training time: the more iterations, the higher the training time. However, compared to Iterative Magnitude Pruning (IMP) we observe that our entropy-based term introduces a negligible overhead (in Table 13, Appendix. D.3, a little bit more than 1 minute per iteration). Including the entropy in the minimized objective function could be a way to design a one-shot approach, which would be more efficient at training time. Nevertheless, this approach is not directly suitable as it relies on a non-differentiable expression and is therefore left as future work.

We acknowledge that our approach should extend to larger models such as large language models to provide insights into its scalability and effectiveness in more complex scenarios. Due to time and resource limitations, we focused our experiments on the tested models. To break the limitation coming from the training cost, we consider replacing full retraining in each iteration with shorter fine-tuning. We performed tests on different models on CIFAR-10 dataset, in which we employed a short fine-tuning process that focused only on the final stage of training. We refer to this approach as NEPENTHE-finetuning. As shown in Table 7, even though the ability of NEPENTHE-finetuning to remove layers and preserve performance is not as remarkable as NEPENTHE. NEPENTHE-finetuning is still functional. This result shows that our method has the potential to be extended to larger language models, and our approach is scalable and effective in more complex situations. Further exploration and refinement of this approach are left for future work.

## 5 CONCLUSION

In this work, we have presented NEPENTHE, an iterative unstructured approach towards layer removal in rectifier-activated deep neural networks. Leveraging on some theoretical results showing that unstructured pruning has the potential to reduce the neural network's depth, an entropy-based weighting mechanism has been designed to select parameters to prune from the network toward depth reduction and attempt to preserve high performance in the considered tasks. Experiments were conducted on popular architectures, including the Transformer-based Swin-T and architectures for NLP like BERT and RoBERTa, showcasing the potential of NEPENTHE to reduce the number of layers in the model concretely. This work has a practical impact even in computation on parallel architectures such as GPUs or TPUs, as it inherently reduces the critical path forward propagation undergoes.

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
