# OpenReview forum: "NEPENTHE: Entropy-Based Pruning as a Neural Network Depth's Reducer"
_ICLR.cc/2025/Conference — ICLR 2025 Conference Withdrawn Submission_

### Official Review · Reviewer_us6w · 2024-11-01

**Soundness:** 2
**Presentation:** 2
**Contribution:** 1
**Rating:** 3
**Confidence:** 5

**Summary:**

The paper proposes a method to perform pruning of trained neural network. The approach is based on per layer entropy and removes entire layers to target a faster model. The paper connects entropy based pruning and weight magnitude pruning, and arrives into a single criteria per layer. The method requires training of the model.

**Strengths:**

- Layer pruning has its benefits such as speeding up the model on GPUs. Other techniques like unstructured pruning does not.
- The approach is relatively simple according to equations, and requires measuring "frequency" of activations, with later normalization.

**Weaknesses:**

- The approach is tested on out-dated models and datasets and as a result is compared with non so recent models.
- Depth pruning is recently studied in LLMs. Most approaches don't require fine-tuning. The proposed approach should be compared to those to demonstrate that the metric is better.

Men, X., Xu, M., Zhang, Q., Wang, B., Lin, H., Lu, Y., Han, X., and Chen, W. Shortgpt: Layers in large language models are more redundant than you expect. arXiv preprint arXiv:2403.03853, 2024.

Kim, B.-K., Kim, G., Kim, T.-H., Castells, T., Choi, S., Shin, J., and Song, H.-K. Shortened llama: A simple depth pruning for large language models. arXiv preprint arXiv:2402.02834, 2024.

Siddiqui, Shoaib Ahmed, Xin Dong, Greg Heinrich, Thomas Breuel, Jan Kautz, David Krueger, and Pavlo Molchanov. "A deeper look at depth pruning of LLMs." arXiv preprint arXiv:2407.16286 (2024).

**Questions:**

For Table 2, did authors reimplemented other techniques for comparison, or took numbers from external source?

---

> ### Author Response · Authors · 2024-11-20
>
> **[W1 - Outdated Models]**
> These models can not be considered outdated as they are the basis for 'modern' models. For example, ResNet18 introduced the revolutionary residual structure, which solved the gradient vanishing problem in deep networks[A7]. This concept is still crucial in deep learning architectures today. BERT transformed NLP by establishing the pretraining-finetuning paradigm, which is still used in state-of-the-art models[A8]. However, we understand the will of the reviewer to confront with some more actual model - we apply NEPENTHE on a pre-trained Llama 3.1-8B, as presented in the answer in the general comment.
>
> **[W2 - Comparison with Depth Pruning in LLMs]**
> Thank you for your suggestion. However, since the key message of our paper is to demonstrate that thanks to unstructured pruning, we can remove entire layers, we compare our approach NEPENTHE with traditional iterative magnitude pruning algorithms, and not with the proposed LLM-tailored approaches.
> Nevertheless, we show that our model-agnostic method NEPENTHE can also be suitable to LLMs in the answer in the general comment.
>
> **[Q1 - Implementation of the Other Techniques]**
> To ensure a fair comparison, all the other techniques were re-implemented and tested following the same training policy described in the Appendix, Table 9 and 10.

---

### Official Review · Reviewer_HV8g · 2024-11-03

**Soundness:** 3
**Presentation:** 3
**Contribution:** 3
**Rating:** 6
**Confidence:** 4

**Summary:**

The paper presents NEPENTHE, an approach aimed at reducing the depth of over-parametrized deep neural networks (DNNs) to decrease computational demands. This method uses an entropy-based strategy to prune layers that exhibit low entropy, allowing for the removal of entire layers from the network with minimal performance loss. The approach is validated on architectures like MobileNet, Swin-T, and RoBERTa.

 NEPENTHE utilizes unstructured pruning that is guided by the entropy of neuron activation within each layer. By re-weighting the pruning process based on entropy levels, the method prioritizes the removal of layers with the lowest entropy. The technique is supported by both theoretical findings and empirical results, showcasing effective reduction of network depth without significant degradation in model performance.

**Strengths:**

Innovation in Pruning: The approach is novel in its use of entropy as a criterion for pruning, shifting the focus from width reduction to depth reduction, which is less explored.
Comprehensive Evaluation: The method is thoroughly tested across several datasets and architectures, providing a robust validation of its effectiveness.
Preservation of Performance: NEPENTHE successfully demonstrates the preservation of model performance even with significant reductions in network depth, addressing a common challenge in neural network pruning.

**Weaknesses:**

Complexity in Implementation: The entropy-based pruning requires careful calibration and may introduce complexity in tuning the parameters for optimal performance across different architectures.
Limited Discussion on Scalability: The method's scalability to extremely large networks.
The function weights_to_prune which includes the whole logic behind the pruning idea is not explained in the main paper.

**Questions:**

Could you provide detailed statistics on the percentage of parameters pruned by the NEPENTHE method compared to other pruning approaches? This information would be valuable for a comprehensive assessment of the method's efficiency and its impact on the overall computational resources required for both training and inference.

How does the removal of different types of layers (e.g., convolutional vs. fully connected) affect the overall network architecture? Are certain layers more "entropy-critical" than others?

How do you choose which samples to use to compute the probability of being on or off of each neuron?

Does the reduced depth network generalize to different dataset as in the lottery ticket hypothesis?

References:
Frankle, Jonathan, and Michael Carbin. "The lottery ticket hypothesis: Finding sparse, trainable neural networks." arXiv preprint arXiv:1803.03635 (2018).

ElAraby, Mostafa, Guy Wolf, and Margarida Carvalho. "OAMIP: optimizing ANN architectures using mixed-integer programming." International Conference on Integration of Constraint Programming, Artificial Intelligence, and Operations Research. Cham: Springer Nature Switzerland, 2023.

---

> ### Author Response · Authors · 2024-11-20
>
> **[W1 - Scalability to extremely large networks]**
> See the answer **[G - Application to more recent architectures (LLM)]** in the general comment.
>
> **[W2 - Function weights_to_prune]**
> This function is discussed in Alg. 2 in the Appendix. We will include it in the main paper to make everything more clear.
>
> **[Q1 - Pruning percentage]**
> We used the same pruning percentage across different pruning approaches. Moreover, we conducted a study on this pruning percentage in Tab. 8 in Appendix Sec. B. We use the current pruning percentage to balance the trade-off between the number of removed layers and performance.
>
> **[Q2 -Entropy-critical layers]**
> Merging a convolutional layer with the subsequent convolution layer results in a layer with larger filters. When merging a fully connected layer with the subsequent layer, we get a layer with dimensions matching the input of the first layer to the output of the second layer for the fusion of two FCs.
>
> Meanwhile, in convolution neural networks, some layers are more “entropy-critical” compared to others. For instance, in ResNet-18, the first layer in the last residual block would be the first to be removed. While in transformers, different layers are removed across different tasks.
>
> **[Q3 - Sample Selection to Compute the Probability]**
> All the samples of the training set are used to compute the probability.
>
> **[Q4 -Generalization of the Reduced Depth to Different Datasets]**
> The reduced depth network does not generalize to different datasets as in the lottery ticket hypothesis. Indeed, as we presented in the answer **[Q2 -Entropy-critical layers]**, for instance in transformers, different layers are removed across different tasks.

---

### Official Review · Reviewer_ECaQ · 2024-11-04

**Soundness:** 2
**Presentation:** 2
**Contribution:** 2
**Rating:** 3
**Confidence:** 4

**Summary:**

The paper presents an iterative pruning method to reduce network depth, based on layer entropy. The layer entropy is based on neuron entropy, which is defined based on the sign of the output activation. They further have an entropy aware pruning score and put it into an iterative pruning algorithm. Experiments on some small-scale datasets on image classification and NLP show the potential merits of the method.

**Strengths:**

1. Not so many pruning papers focuses on reducing depth of deep models. Depth reduction can bring significant speedup.

2. The idea of using entropy for unstructured pruning sounds interesting, esp. the study of the effect of pruning on layer entropy (Sec. 4.2).

**Weaknesses:**

1. Methodologically, the method may have clear flaws.
- The entropy of a neuron is defined based on its sign. I am not sure if this is grounded. For ReLU networks, the neuron's output is nearly always positive, then the entropy is 0, and thus can be removed according to this paper. This is clearly not correct.
- L186: "The output of the i-th neuron is always the same as its input, this neuron can in principle be absorbed by the following layer as there is no non-linearity between them anymore." -- this statement is also baseless. It only applies to ReLu networks, while for networks with other activation functions (such as sigmoid), even when the output of a neuron is always positive, the nonlinearity cannot be omitted.
- L198 - 205, the method derivation is based on weight Gaussian and input Gaussian assumptions. I am not sure they really hold in today's practical models.


2. As pointed in the paper, the method needs iterative pruning which is very costly. And this invites the comparison fairness problem. For many pruning methods, they do not need so many training epochs. And we know the training epochs will impact the performance significantly. The question is, if the other methods, equipped with the same long training epochs, will they just compete the proposed method?


3. Experiments
- Some of the results look strange. Tab. 1, why resnet18 on cifar10 only has 91.66% top1 accuracy? ResNet56 can easily get to 93.5% and ResNet18 is designed for ImageNet, which has many more params than ResNet56.
- Most of the compared methods are baseline approaches, probably implemented by the authors, lacking comparison with more recent papers.


4. (presentation, writing) issues and questions
- Eq 2, missing punctuation.
- One closely related work is missing - https://proceedings.neurips.cc/paper/2016/hash/41bfd20a38bb1b0bec75acf0845530a7-Abstract.html
- the function Weights to prune -> The function
- this neuron can in principle be absorbed by -> This

**Questions:**

Are there any results on ImageNet-1K with Resnet50 and a ViT model like Vit-B/16?

---

> ### Author Response · Authors · 2024-11-20
>
> Many thanks for your comments. We respectfully disagree with your claim for “clear flaws” — and here below is a rationale behind which we believe there was a misunderstanding by the reviewer. We agree in general that we should be in some parts more precise, and we commit to providing an updated version of the paper in such spirit.
>
> **[W1 - Entropy of a Neuron & Removing Neurons Whose Output is Always Positive]**
> We do not estimate the entropy of the information but rather the entropy as defined in [A2].
> The method to merge a neuron whose output is always positive with the next layer by linear combination is similar to [A3].
>
> **[W2 - NEPENTHE Only Applies to ReLU]**
> NEPENTHE is not bound to any rectifier and works with any. Indeed, as shown in Table 6, we also validated NEPENTHE on SiLU, PReLU, LeakyReLU, and GELU-activated networks. For non-rectifier activations like sigmoid, NEPENTHE needs to be adjusted to take into account a different number of states.
>
> For instance, the sigmoid can be divided into three states: 'OFF', 'always output one', and 'output approximates input plus 0.5'. When the entropy equals 0, the neurons that only use the first state can simply be pruned, while for the neurons in the two other states, they can be merged with the next layer by linear combination.
>
> **[W3 - Gaussian Assumptions]**
> With modern weight initialization methods like those presented in [A4] and [A5], the weights are under Gaussian distribution. The input Gaussian assumption is stated in [A6].
>
> **[W4 - Training Epochs]**
> The compute time for NEPENTHE and IMP are presented in Appendix C.3 (Tab. 13 to Tab. 33). These results highlight that compared to other methods, NEPENTHE can either remove more layers or maintain better performance with the same training time.
>
> **[W5 - Performance Difference of ResNet-18 on CIFAR-10]**
> The difference in performance can be caused by various factors: the seed, the library versions, or a difference in the training policies. However, to ensure a fair comparison with all the baselines, we conducted the experiments using the same seed, library versions, and training policies, which can be found in the Appendix.
>
> **[Q1 - More Comparison]**
> Thank you for your suggestion. However, since the key message of our paper is to demonstrate that thanks to unstructured pruning, we can remove entire layers, we compare our approach NEPENTHE with traditional iterative magnitude pruning algorithms, and not with the proposed structured pruning baselines.

---

> > ### Comment · Reviewer_ECaQ · 2024-11-25
> > **Rebuttal reviewed, score maintained.**
> >
> > Thanks for the authors' rebuttal!
> >
> > [W1] The authors responded: "*We do not estimate the entropy of the information but rather the entropy as defined in [A2]. The method to merge a neuron whose output is always positive with the next layer by linear combination is similar to [A3]*." -- Thanks for clarifying this. After checking [A2], I found this work heavily builds upon [A2]. [A2] tries to use unstructured pruning to remove depth, which is exactly what this work is doing. Then, what is the novelty of this work?
> >
> > [W2 - NEPENTHE Only Applies to ReLU] "NEPENTHE needs to be adjusted to take into account a different number of states." ... "For instance, the sigmoid can be divided into **three** states" -- This is exactly why I said the method looks ungrounded. Why the Sigmoid should be divided into three states? Is this just an intuitive division? The proposed method essentially discretizes the continuous output into several discrete ranges and calculates entropy based on that. Have the authors verified the validness of this discretization-based method? If we choose different discretization intervals, will the ranking based on entropy change? I guess there are too many questions here and not solvable in this rebuttal. The theoretical ground of the method needs a lot of work to rebuild.
> >
> > [W3 - Gaussian Assumptions] Citing existing works does not lend more legitimacy to *this* work. The fact that others have that assumption does not mean you can do the same without any practical sanity checks.
> >
> > [W5 - Performance Difference of ResNet-18 on CIFAR-10] Using the same training configs does not mean the experiment is valid. If the config is wrong, then all the methods are compared under the same wrong configs. It is fair, but no any valid conclusions can be drawn from these experiments. The authors do not provide more evidence to show their experiments are convincing.
> >
> > [Q1 - More Comparison] I feel confused. This work focuses on removing layers, which should fall into the structured pruning group. Then why the authors think they should *not* compare with the "structured pruning baselines" and only compare to the IMP baselines?
> >
> > All in all, the paper clearly has too many problems to be fixed in its current shape. I thereby maintain my score.

---

> > > ### Author Response · Authors · 2024-11-28
> > >
> > > Thank you for your detailed feedback. We are happy to explain further here.
> > >
> > > **[Difference between our work and A2]**
> > > The main differences between our approach and [A2] are as follows:
> > >
> > > 1：Better Entropy Calculation:
> > > In our method, we exclude the value '0' when calculating the entropy of activation functions. This prevents the calculation from being influenced by neurons that always output 0 or positive values. In  [A2], neurons with outputs at or above 0 could still appear to have non-zero entropy, which might give less accurate results.
> > >
> > > 2：Focus on Individual Neurons:
> > > Unlike [A2], which prunes entire layers without looking closely at individual neurons, our method is more precise. We focus on pruning neurons with non-zero entropy. This makes the pruning process more effective and easier to understand.
> > >
> > > 3：Stronger Theoretical Support: We provide a clear explanation of how unstructured pruning in layers with rectifier activation functions naturally reduces the layer’s entropy (explained in Sec. 3.1 and Appendix A). This theoretical grounding strengthens our approach compared to [A2].
> > >
> > > Thanks to these differences, our method achieves **better performance** for the same number of layers removed, or with even more layers removed.
> > >
> > > **[Gaussian Assumptions]**
> > > In the revised version, we include in Appendix B, the detailed practical sanity checks for the weights distributions and the inputs distributions of all the considered layers for each architecture employed in our experiments. Looking at Figures 4 to 13, it appears that the weights and the inputs of the corresponding layers follow a Gaussian distribution, validating our initial assumption.
> > >
> > > **[Performance Difference of ResNet-18 on CIFAR-10]**
> > > We respectfully disagree with the assertion that the used configuration is wrong.  The training configuration we employed is based on well-established practices in the field, as demonstrated by its adoption in multiple relevant papers[Z1, Z2, Z3]. This configuration is commonly accepted for experiments on CIFAR-10, and we believe it provides a fair and consistent basis for comparison. While we acknowledge that no configuration is perfect, the setup we used is based on widely recognized standards, and we are confident that our conclusions are valid within this framework.
> > >
> > > [Z1] : Zheng He, Zeke Xie, Quanzhi Zhu, Zengchang Qin. Sparse Double Descent: Where Network Pruning Aggravates Overfitting, ICML 2022.
> > >
> > > [Z2]: Jonathan Frankle, Michael Carbin. The Lottery Ticket Hypothesis: Finding Sparse, Trainable Neural Networks, ICLR 2019.
> > >
> > > [Z3]: Kaiming He, Xiangyu Zhang, Shaoqing Ren, Jian Sun. Deep Residual Learning for Image Recognition, CVPR 2016.
> > >
> > > **[Q1 - More Comparison]**
> > > Related to more comparisons, we also care about the energy consumption that we would introduce in doing more experiments. We consider the comparisons asked not to add sufficient value to the paper (our point is to show that unstructured pruning can also induce layer pruning, which we succeeded in showing). We are open to providing more experiments with comparison to other existing techniques that induce layer pruning by unstructured sparsity (if they exist- which is not the case to our knowledge).

---

### Official Review · Reviewer_hFP2 · 2024-11-05

**Soundness:** 3
**Presentation:** 3
**Contribution:** 3
**Rating:** 3
**Confidence:** 4

**Summary:**

This paper presents NEPENTHE, a pruning method designed to reduce the depth of over-parameterized deep neural networks, addressing their computational inefficiency in real-time applications. Unlike traditional pruning that focuses on reducing model width, NEPENTHE uses an entropy-based unstructured pruning technique to target and remove entire low-entropy layers, effectively shrinking the network's depth without compromising performance. NEPENTHE is validated on models like MobileNet, Swin-T, and RoBERTa and demonstrates that certain layers can be linearized when networks are over-parameterized, leading to reduced computational demands while retaining accuracy.

**Strengths:**

Quality and Clarity
- The paper is clearly written, with sufficient background and motivation for the problem

Originality and Significance
- The proposed iterative entropy guided pruning scheme is novel

Experiments
- The paper studies the generalizability of the method across wide variety of models and applications ranging from a Resnet for classification to a RoBERTa for language tasks
- The paper provides thorough details on the experimental setup (in the appendix)
- The analyses provided on neuron states per layer, allowing to drop entire layers (hence allowing for some structured sparsity and latency gains) is interesting and valuable contribution
- The ablation study provided in Table 4 depict the value of each component of NEPENTHE

**Weaknesses:**

- I find the model types studied to to limited. Since overparameterization is most prelevant in modern decoder-only language models, I think studying this approach eg: on a Llama-3.1-8B and larger scales would greatly improve the applicability and impact of the work
- Baselines: The paper mainly compares against Iterative magnitude pruning and comparison with more recent pruning methods is missing [1,2,3,4] and [5,6,7] for large language models. I encourage the authors to also compare to different baselines in terms of compute time taken for pruning and also compare against zero-shot pruning methods (after recovery fine-tuning).
- Given that the method relies on expensive training after every iteration of weight pruning, this makes adopting the method to much larger scales inefficient. Could the authors comment on the scalability of the method?
- Latency gains in practice: Given that NEPENTHE does not guarantee completely sparse prunable layers, hence structured pruning. I find the practical latency gains of the method quite limited, if they cannot be enforced/specified by the user.

[1] Kurtic, E., Campos, D., Nguyen, T., Frantar, E., Kurtz, M., Fineran, B., Goin, M. and Alistarh, D., 2022. The optimal bert surgeon: Scalable and accurate second-order pruning for large language models. arXiv preprint arXiv:2203.0725

[2] Yu, F., Huang, K., Wang, M., Cheng, Y., Chu, W. and Cui, L., 2022, June. Width & depth pruning for vision transformers. In Proceedings of the AAAI Conference on Artificial Intelligence (Vol. 36, No. 3, pp. 3143-3151)

[3] Ilhan, F., Su, G., Tekin, S.F., Huang, T., Hu, S. and Liu, L., 2024. Resource-Efficient Transformer Pruning for Finetuning of Large Models. In Proceedings of the IEEE/CVF Conference on Computer Vision and Pattern Recognition (pp. 16206-16215).

[4] Klein, A., Golebiowski, J., Ma, X., Perrone, V. and Archambeau, C., 2024. Structural Pruning of Pre-trained Language Models via Neural Architecture Search. arXiv preprint arXiv:2405.02267.

[5] Sun, M., Liu, Z., Bair, A. and Kolter, J.Z., 2023. A simple and effective pruning approach for large language models. arXiv preprint arXiv:2306.11695.

[6] Ashkboos, S., Croci, M.L., Nascimento, M.G.D., Hoefler, T. and Hensman, J., 2024. Slicegpt: Compress large language models by deleting rows and columns. arXiv preprint arXiv:2401.15024.

[7] Ma, X., Fang, G. and Wang, X., 2023. Llm-pruner: On the structural pruning of large language models. Advances in neural information processing systems, 36, pp.21702-21720.

**Questions:**

Questions
- Check weaknesses
- Did the authors observe any particular trends in the layer types that are typically dropped or have larger number of neurons set to "off" state. Are the observations different across model families?
- How does the compute time for NEPENTHE compare to different baselines?
- Given that perplexity often doesn't translate to donwstream task performance (in-context learning properties) for decoder-only LLMs [1]. How do the authors see NEPENTHE being extended to large language models? Given that NEPENTHE requires iterative training, could the authors comment on the scalability of the approach?

[1] Jaiswal, A., Gan, Z., Du, X., Zhang, B., Wang, Z. and Yang, Y., 2023. Compressing llms: The truth is rarely pure and never simple. arXiv preprint arXiv:2310.01382.

---

> ### Author Response · Authors · 2024-11-20
>
> **[W1 - Application to Modern Large Language Models]**
> See the answer **[G - Application to more recent architectures (LLM)]**.
>
> **[W2 - More Baselines]**
> Thank you for your suggestion. However, since the key message of our paper is to demonstrate that thanks to unstructured pruning, we can remove entire layers, we compare our approach NEPENTHE with traditional iterative magnitude pruning algorithms, and not with the proposed structured pruning baselines as their focus is different.
>
> **[W3 - Scalability]**
> As we discussed in Section 4.5, lines 518-525, by applying NEPENTHE with a short finetuning strategy instead of full retraining, our pruning method has the potential to be extended to larger-scale tasks as demonstrated by **[G - Application to more recent architectures (LLM)]** in the general comment. Further exploration and refinement of finetuning are left for future work.
>
> **[W4 - Latency Gains]**
> A layer with zero entropy can be merged with the next layer through a linear combination, effectively removing the layer. Hence, as demonstrated in Table 5 in Section 4.5, this approach leads to latency gains in practice.
>
> **[Q1 - Trends in the Removed Layers]**
> On the one hand, in convolutional neural networks, the same layers tend to be discarded. For instance, in ResNet-18, the first layer in the last residual block would be the first to be removed. While in transformers, different layers are removed across different tasks.
>
> **[Q2 - Compute Time]**
> The compute time for NEPENTHE and IMP are presented in Appendix C.3 (Tab. 13 to Tab. 33). These results highlight that compared to other methods, NEPENTHE can either remove more layers or maintain better performance with the same training time.
>
> **[Q3 - NEPENTHE Being Extended to Large Language Models]**
> See the answer [G - Application to more recent architectures (LLM)] in the general comment.

---

> > ### Comment · Reviewer_hFP2 · 2024-11-25
> >
> > I thank the authors for answering my questions. However, concerns raised about the baselines used in this work and a more thorough investigation of NEPENTHE were not answered satisfactorily. I maintain my score

---

> > > ### Author Response · Authors · 2024-11-25
> > > **Could you be more precise?**
> > >
> > > Dear reviewer,
> > >
> > > Related to the baseline, please notice that **we have provided in the general comment section an experiment on Llama-3.1-8B**. The results obtained on this architecture confirm the paper's observations, and further validate our baseline selection that, although being less recent than other models, they embody their essential building blocks.
> > >
> > > Related to **more comparisons**, we also care about the energy consumption that we would introduce in doing more experiments. We consider the comparisons asked not to add sufficient value to the paper (our point is to show that unstructured pruning can induce also layer pruning, which we succeeded in showing). We are open to providing more experiments on techniques that induce layer pruning by unstructured sparsity (if they exist- which is not the case to our knowledge).
> > >
> > > Finally, related to the **more thorough investigation**, we invite the reviewer to follow the pointers already provided in the original rebuttal.
> > >
> > > We are very open to further discussing these points. Thank you.

---

### Author Response · Authors · 2024-11-20

Dear reviewers,

Many thanks for your invaluable punctual insights, which will improve the next version of the paper. Below you can find one general comment, while more specific ones will be provided for each reviewer, for which all the references are provided here as well.

**[G - Application to more recent architectures (LLM)]**

Due to the lack of computing resources and keeping in mind energy consumption, we focused our experiments on traditional image classification as well as natural language processing setups, which in our belief are sufficient to showcase the core contribution of our paper, being that unstructured pruning can lead to layer collapse. However, to showcase the applicability and the impact of our work, we apply NEPENTHE on a pre-trained Llama 3.1-8B (as suggested by rev. **hFP2**): we prune it iteratively without any fine-tuning, setting $\zeta = 0.25$.

The results for Llama 3.1-8B (16FP) evaluated on MMLU - High school US history are presented in the table below. Even without fine-tuning, NEPENTHE can remove 7 layers and perform better than the dense model (on the test set). After 9 removed layers, the performance drops significantly.

| Val Acc | Test Acc | Lay. Removeable |
|:-------:|:--------:|:---------------:|
|  40,91  |   33,82  |        0        |
|  38,49  |   36,76  |        3        |
|  36,36  |   29,9   |        5        |
|  33,82  |   35,08  |        7        |
|  31,82  |   28,92  |        9        |
|  18,18  |   27,45  |        11       |

Table: The accuracy and number of layers removable for Llama 3.1-8B(16FP) evaluated on  MMLU - High school US history and pruned by NEPENTHE without fine-tuning.

This experiment shows that our approach NEPENTHE can be suitable for modern decoder-only language models since it enables the removal of entire layers in Llama models.

We believe that applying a healing policy after pruning might be beneficial in recovering performance, as shown in [A1].

**References**

[A1] Chen, T., Ding, T., Yadav, B., Zharkov, I., and Liang, L. (2023). Lorashear: Efficient large language model structured pruning and knowledge recovery. arXiv preprint arXiv:2310.18356.

[A2] Liao, Z., Quétu, V., Nguyen, V. T., and Tartaglione, E. (2023). Can Unstructured Pruning Reduce the Depth in Deep Neural Networks?. In Proceedings of the IEEE/CVF International Conference on Computer Vision (pp. 1402-1406).

[A3] Pilo, G., Hezbri, N., Pereira e Ferreira, A., Quétu, V., and Tartaglione, E. Layerfold: A Python Library to Reduce the Depth of Neural Networks. [Publication details pending].

[A4] Glorot, X., and Bengio, Y. (2010). Understanding the difficulty of training deep feedforward neural networks. In Proceedings of the Thirteenth International Conference on Artificial Intelligence and Statistics (pp. 249-256).

[A5] He, K., Zhang, X., Ren, S., and Sun, J. (2015). Delving deep into rectifiers: Surpassing human-level performance on ImageNet classification. In Proceedings of the IEEE International Conference on Computer Vision (pp. 1026-1034).

[A6] Neal, R. M. (1996). Priors for infinite networks. In Bayesian Learning for Neural Networks (pp. 29-53).

[A7] He, K., Zhang, X., Ren, S., and Sun, J. (2016). Deep residual learning for image recognition. In Proceedings of the IEEE Conference on Computer Vision and Pattern Recognition (pp. 770-778).

[A8] Devlin, J., Chang, M.-W., Lee, K., and Toutanova, L. (2019). BERT: Pre-training of deep bidirectional transformers for language understanding. In Proceedings of the 2019 Conference of the North American Chapter of the Association for Computational Linguistics: Human Language Technologies (pp. 4171-4186).

---

### Comment · Area_Chair_DuzM · 2024-11-22
**Discussion**

Dear reviewers,

The authors have responded to your reviews.

Until November 26th @ 2359 (AOE time) reviewers and authors can freely exchange responses, so if there any clarifications you require from the authors, now is the time to seek them!

Best,

AC

---

### Author Response · Authors · 2024-11-25
****Solicitation to interact after rebuttal****

Dear Reviewers,

We strongly believe we have addressed your concerns and we are happy to further discuss with you - we feel the current evaluation no longer reflects the value of the paper after our rebuttal. If there is no further comment form your side, we ask you to adequate your evaluation of the paper.
Thank you,

The Authors

---

### Note · Authors · 2024-12-25

I have read and agree with the venue's withdrawal policy on behalf of myself and my co-authors.